# ElectroMagnetic Music: a new tool for attracting people interest in Geosciences, while sensitizing them to planet sustainability

Antonio Menghini[1], Stefano Pontani[2], Vincenzo Sapia[3], Tiziana Lanza[3]

[1] Aarhus Geofisica s.r.l., Via Giuntini 13, 56021, Cascina (PI), Italy

[2] Musician, Via Treviso 15, Viterbo, Italy

[3] Istituto Nazionale di Geofisica e Vulcanologia, Via di Vigna Murata 605, 00143, Roma, Italy

*Corresponding author*: Antonio Menghini (am@aarhusgeo.com)

**Abstract.** In recent years, different sonification methods used to organize scientific work have come out of the scientific realm to cross into other areas and achieve purposes other than those pursued strictly by scientific research. The ElectroMagnetic Music (EMusic), a project born in Italy, fits fully into this area. By transforming into musical pitches the voltage response collected by Transient ElectroMagnetic Method (TEM), a well-known geophysical tool for subsurface exploration, this novel approach enables to extract musical pieces reflecting the effective geological setting and providing its own soundtrack (i.e. the "soundscape," the audio component of a landscape). The soundscape becomes the basis where a dedicated band improvises jazz music. Besides being a new method for creating music, the project has not only the ambitious goal to attract people's interest on Earth sciences and prospecting methods, but also raising awareness of the environmental problems that characterize geological sites through the music. In this work, we explore into the EMusic experiences gained as a live band around the world. We also report some preliminary data on people reaction and anticipate some future plans for better assessing the potential of the method as a good science communication tool.

## 1 Introduction

Music is powerful in sensitizing people, thanks to its capacity to involve everybody, without barriers of language, culture and religion. Most recently, sonification of scientific data is becoming more and more popular, obtaining great visibility in the media. Just to mention some recent approaches: sonification of gravitational waves (Hughes, 2016), planet orbits (Quinton and Benyon, 2016), global temperature variations (Hilgren, 2019), seismic noise (Avanzo et al., 2010), earthquakes (Michael, 2013) and geophysical data (Dell'Aversana et al., 2016). In this trend, the EMusic is the first project that, utilizing the ElectroMagnetic (EM) responses of the Earth, provided a new method to sonify data strictly related to the geological nature of the subsoil. From the available literature around, sonification is the use of non-speech audio to convey information or to "feel" data. In this term, sonification turns to representational techniques in which data sets or selected features of a specific output signal are mapped into audio signals. This concept has an old origin since it was exploited for capturing the main structure of

complex data clustering in a way of being more discernible for humans. To date, the sonification becomes almost an immediate tool to perceive some physical parameters. For instance, it is widely used in medicine for monitoring patient health parameters, for detecting the level of radioactive concentration through Geiger counter or the type of buried ferrous/metal objects by means of a metal detector among many others. All these examples could then be reasonably considered as a process of sonification in which we are capable of translating specific flow of data derived from a physical excitation of natural materials, by means of external source of energy, into audio signals. Therefore, we are able to identify something that we cannot directly see by listening to its signal.

Sonification is different from audification. With the second term, we identify an auditory display technique for representing a sequence of data values as sound. By definition, it is described as a "direct translation of a data waveform to the audible domain" (Roger, 2009), so that it typically requires large data sets with periodic components. Nevertheless, both sonification and audification are representational techniques in which data sets are turned into audio signals. Both are appropriate tool to perceive when the information being displayed changes in time (i.e. climate change,) or intensity (i.e. health parameters).

However, their relationship can be demonstrated in the way data values in some sonifications that directly define audio signals are called audification. From Hermann and Ritter 1999, we can understand also how sonification is a good tool for rapid screening of complex clustering data so that it has a higher potential for interacting with the raw records, as a preliminary step, also in conjunction with their visualization.

Anyway, a detailed overview of sonification in general and the status of the research in this specific field is beyond the scope of this paper so, for an in-depth discussion of sonification and audification concepts, the reader can refer to many other already published papers (such as Kramer et al., 2010; Walker and Nees, 2011; Mora et al., 2020).

In recent years both these techniques have come out of the scientific realm, to cross into other areas and achieve purposes other than those previously described. Sonification is used often in the artistic sphere as a valuable vehicle for bringing science to the general public. We set out to use EMusic not only to excite people and to have fun with musicians, but also to bring people closer to earth sciences and their investigative methods.

To do this, we directly investigate down to the subsurface, looking at specific physical property of the Earth. Thus, it is not out of place to claim that we can extract the effective "Sound of the Earth". In fact, the source of our sonification process depends upon the electrical behaviour of the rocks (i.e. the resistivity).

Similar to other sonification process, we adopt a mathematical rule that allows us to translate the geophysical data into audible frequencies, following a procedure codified by Menghini and Pontani (2016). It would be therefore more correct to say that we can produce pitches, rather than sounds. The musical notes can be played by any kind of instrument, also by a human voice. The involvement of the musicians is direct, as they have to arrange an improvisation or a composition, by using these pitches provided by the Earth: the EMusic data are the bricks that will be used to build any musical performance (some examples have been reported by Menghini, 2016, 2018 and Duncombe, 2019). This allows to achieve an effective connection between Art (Music) and Science (Geology), in a way that can be easily appreciated by non-expert audience. Geoscientists actively

participate in the composition of the work, so that they can be considered as composers, or rather as the medium between the Earth and the performers, also by providing some keywords on the history of the geological site inspiring the musicians.

65

## 2 Objectives

The EMusic has been conceived to translate into "music" data acquired by a specific scientific instrument, normally used for many geoscience applications. We believe that this technique has a great potential in terms of science and art communication capability. To get a first taste of these potentials, in a first phase, our agenda included mainly live events in several geo-sites. We performed all around the world in close cooperation with musicians to promote the EMusic. We also used the net to spread our method of sonification, the events performed, and the ones scheduled. In the near future, we intend to bring the project in schools to involve students in Earth sciences, planet sustainability while introducing them to a different approach to music.

For the time being, as a live band, we are satisfied since the project obtained great interest by the scientific and musical communities. The EGU General Assembly in Wien invited us to play twice (2017 and 2018). Geoscience Australia invited us to play in Canberra and Perth; AGU Centennial Grant awarded us with a 5 hours sound installation based on Airborne EM data collected in Colorado Mountains; Under the patronage of the City of Naples, the Geological Survey of the Campania District and in collaboration with the National Park of Vesuvius, we played on the top of Vesuvius Volcano; We also performed at the INGV (Istituto Nazionale di Geofisica e Vulcanologia) Open Day; we carried out a tour of 7 stages "Sounds from the Geology of Italy", based on the sonification of EM data collected in some of the most beautiful natural and cultural sites, involving famous international jazzists, like Enrico Rava and Francesco Cafiso.

This paper describes in details our method of sonification and refers on the events performed in collaboration with INGV. We describe also the potentialities of the methods from a science communication perspective even if so far we did not conduct an evaluation. Nevertheless, as previously said, we preview in a near future to experiment the method in schools to have the opportunity to extensively evaluate its efficaciousness in terms of attracting students' interest in geosciences while sensitizing them to planet sustainability. To this aim we are at present implementing a project named Georisonanze (Georesonances).

## 3 Methods

### 3.1 TEM method

Before describing our sonification method, we dedicate a paragraph to the geophysical method at the basis of it. The Transient EM method (TEM) is a well-known geophysical technique, widely applied in the field of geoscience, since the 50s, to detect mining and groundwater resources (Fitterman et al., 1986). TEM measurements can be performed directly on the ground or through a sophisticated airborne system (AEM hereinafter). The main difference is that AEM is able to map wide areas in a relatively short time. The collected TEM data can be used in the construction of accurate three-dimensional geological models

(Sapia et al., 2015), after careful data processing and system calibration (Viezzoli et al., 2013; Sapia et al., 2015). Nevertheless, ground TEM data is a valuable tool to map subsurface structure in many field of applications, from geological mapping of seismic areas (Civico et al., 2017; Villani et al., 2019) to groundwater studies (Auken et al., 2003) although the recovered models differ in terms of structure resolutions compared to AEM ones as due to a limited number of measured field data (Sapia et al., 2014). The electromagnetic induction theory, which is at the base of TEM methods, is well described from Nabighian

and Macnae (1991). In general, the instrumentation is characterized by two main components, a transmitter square loop and a receiver induction coil. A current is forced to flow in the loop thus generating a magnetic field, which is stable inside and outside the loop unless no current changes occur. The current is then suddenly turned off and this generates a change in the magnetic field (i.e. rapid magnetic flux variation) which, in turn, induces a current to flow in the ground (Figure 1). The induction currents (known as eddy current) diffuse downward and outward with a decreasing rate which strictly depend on the

electrical properties of the medium (i.e. its resistivity). The higher the resistivity the faster is the current decay and the lower is the signal strength (voltage). The receiver coil (Figure 2) records the rate of change of the magnetic field produced by the decay of the eddy currents, by means of several time gates, which are specifically designed to accurately catch the transient curve in the form of voltage values over time (Figure 1c).

  Therefore, the sampling of this response is achieved by means of a series of gates having different widths, which increase with

time as the signal becomes weaker and weaker going into depth. Since the voltage response for each time gate is assigned to its centre, it follows that the data are spaced so that they become more and more distant from each other.

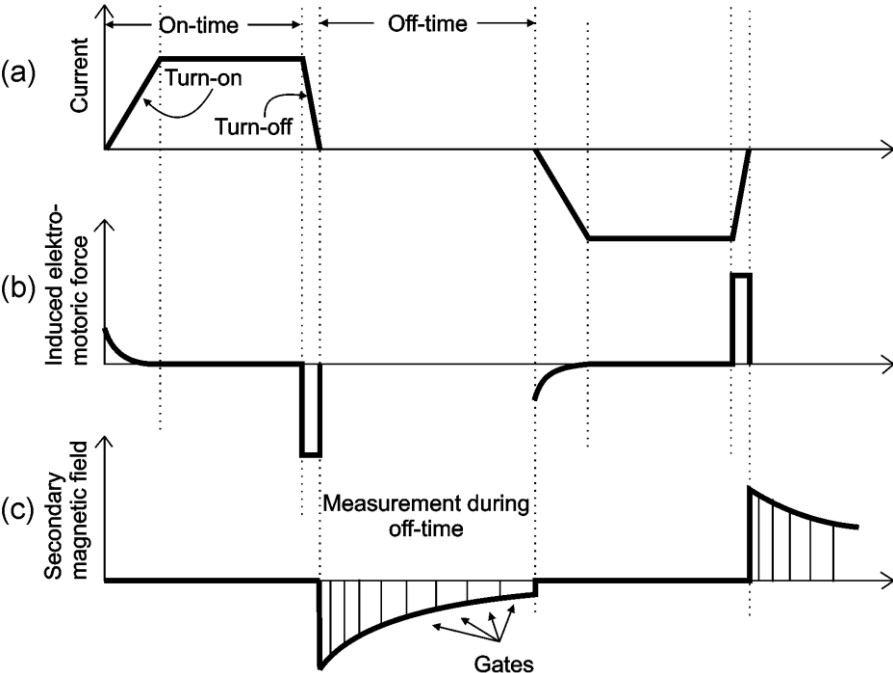

**Figure 1: Current injected in the transmitter loop (a). The induced electro-motoric force in the ground (b). The secondary magnetic field measured in the receiver coil (c). The "listening time" is during the off-time. (adapted from Christiansen et al., 2006).**

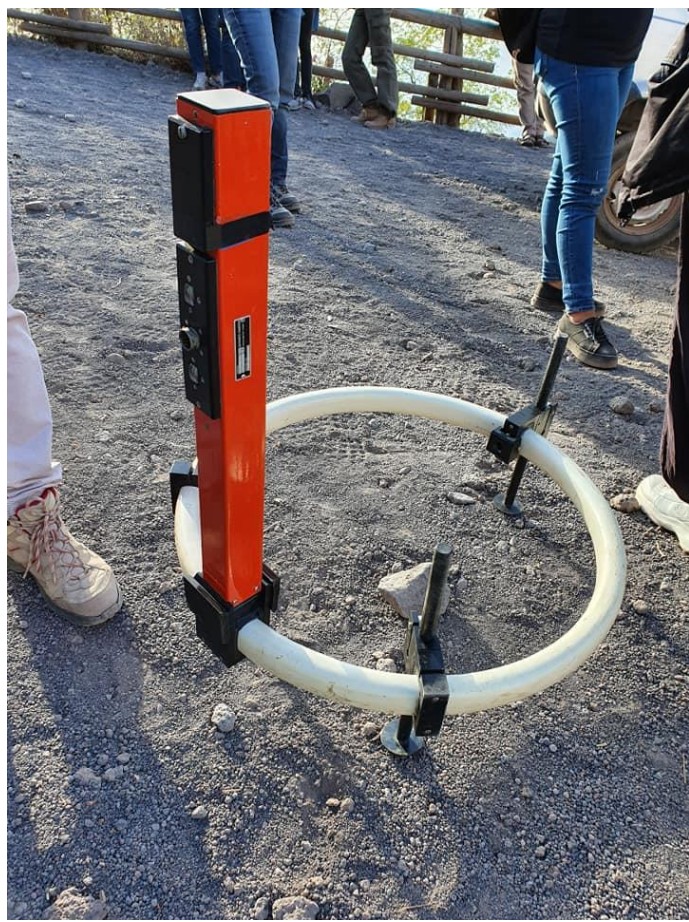


**Figure 2: The receiver coil that records the Earth EM response.**

### 3.2 Data sonification

The "listening time", that we use to sonify the Earth response, regards the period in which the system is off (Figure 1c) and no
currents are flowing into the wire (off-time).

Regarding the sonification's rules, we need to start by considering that the sound frequency associated to musical pitches are distributed in a non-linear scale (according $\log_2$). Tab. 1 shows the relationship between frequency and pitch, according to the equal tempered scale (twelve note scale).

| | C | C# | D | Eb | E | F | F# | G | G# | A | Bb | B |
|---|---|---|---|---|---|---|---|---|---|---|---|---|
| 0 | 16.35 | 17.32 | 18.35 | 19.45 | 20.60 | 21.83 | 23.12 | 24.50 | 25.96 | 27.50 | 29.14 | 30.87 |
| 1 | 32.70 | 34.65 | 36.71 | 38.89 | 41.20 | 43.65 | 46.25 | 49.00 | 51.91 | 55.00 | 58.27 | 61.74 |
| 2 | 65.41 | 69.30 | 73.42 | 77.78 | 82.41 | 87.31 | 92.50 | 98.00 | 103.8 | 110.0 | 116.5 | 123.5 |
| 3 | 130.8 | 138.6 | 146.8 | 155.6 | 164.8 | 174.6 | 185.0 | 196.0 | 207.7 | 220.0 | 233.1 | 246.9 |
| 4 | 261.6 | 277.2 | 293.7 | 311.1 | 329.6 | 349.2 | 370.0 | 392.0 | 415.3 | 440.0 | 466.2 | 493.9 |
| 5 | 523.3 | 554.4 | 587.3 | 622.3 | 659.3 | 698.5 | 740.0 | 784.0 | 830.6 | 880.0 | 932.3 | 987.8 |
| 6 | 1047 | 1109 | 1175 | 1245 | 1319 | 1397 | 1480 | 1568 | 1661 | 1760 | 1865 | 1976 |
| 7 | 2093 | 2217 | 2349 | 2489 | 2637 | 2794 | 2960 | 3136 | 3322 | 3520 | 3729 | 3951 |
| 8 | 4186 | 4435 | 4699 | 4978 | 5274 | 5588 | 5920 | 6272 | 6645 | 7040 | 7459 | 7902 |

**Tab. 1: Relationship between frequencies and pitches.**

The Musical Instrument Digital Interface (MIDI) pitch values are linked with frequency through the following equations:

$f = 440 * 2^{(m-69)/12}$ (1)

$m = 69 + 12 \log_2 (f/440)$ (2)

where f = frequency and m = MIDI pitch

In order to listen to the sonified voltage response, we need to normalize the geophysical values so that to stay within the MIDI range between 0 and 127 units, that correspond to the frequency range between 8,176 and 12543,854 Hz. Actually, the audible range is limited to 10-110 MIDI units. To convert geophysical data into musical notes, the EMusic exploits this simple formula:

$X_N = 10 + [Log (X/X_{MIN}) * 100/Log (X_{MAX}/X_{MIN})]$

Where

$X_N$ = normalized value

X = measured voltage value

$X_{MAX}$ = maximum voltage value

$X_{MIN}$ = minimum voltage value

The choice of $X_{MAX}$ and $X_{MIN}$ is a key-point and it was achieved after having considered a wide statistical assessment of EM datasets collected all over the World, hence in very different geological contexts. The highest limit can be assigned to seawater (to which corresponds a resistivity of 0.2-0.3 ohm-m), while the lowest one can be measured for highly resistive rocks (higher than 1000 ohm-m). In order to make comparable any sonified transients, we prefer to use the same range of voltage, i.e. fixed

values for minimum and maximum response: this device allows to compare different geological scenario and different EM systems in an objective way. This approach can be similar to the choice of the edges of a colour scale used for imaging a

physical parameter (e.g. resistivity). Of course, this means that we cannot exploit the full frequency range for any sounding: in the case of a highly conductive situation, we could stay within a narrow sequence of high tones, while, for very resistive environment, we could get only low tones separated by wide intervals. It follows that anyone can immediately understand in

which geological scenario we are, simply by hearing the EMusic outcome. Based on these considerations, the values were fixed as $X_{MAX} = 1$ E-3 V/Am$^4$ and $X_{MIN} = 1$ E-12 V/Am$^4$.

Once the value of $X_N$ is obtained, we assign for each sampling gate the closest pitch of Tab. 1, as we prefer to adopt the equal tempered scale, so that to make the music more appealing (at least for a standard Western listener). An alternative approach could be the use of microtones, hence musical notes that are not in the twelve-note scale: in this case any precise frequency

drawn from the voltage response is used as it is, without any approximation. But, in this case, the musical performance can be carried out only by means of electronic instruments or by acoustic instruments with different tuning.

Figure 3 shows a typical transient (i.e. the voltage response of the Earth, expressed as V/m$^4$A) as drawn from the field data reported in Tab. 2. The gates are figured by using a linear scale for the time. As the transient works out within a few milliseconds, we adopted a time expansion that can be chosen depending on how long we want to arrange the composition

(usually we use values between 100,000 and 1 million). Otherwise our ear would hear a single chord formed by all the gates/pitches. Indeed, in the so-called "Flight Mode" we used this approach, as we have to handle several soundings collected during an Airborne EM survey: in this particular kind of prospection, the data are acquired during a flight, so that huge amount of soundings can be sonified. An example was presented at the EGU 2017 Assembly (Menghini and Pontani, 2017) where a flight mode composition was arranged by using AEM data collected over Sierra Leone.

On the contrary, due to the large dynamic range, the voltages are shown in log scale. Figure 3 refers to an Airborne TEM sounding collected over Selinunte temple, in Sicily and it was the first track of the concert we performed at EGU Assembly in 2018. For each gate we reported the corresponding pitches that were used by the musicians to improvise.

| Voltage (V/Am$^4$) | MIDI | Frequency |
|---|---|---|
| | | |
| 3.7E-09 | 101.364 | 2.85E+03 |
| 3.22E-09 | 100.1571 | 2.66E+03 |
| 2.71E-09 | 98.65939 | 2.44E+03 |
| 2.38E-09 | 97.53154 | 2.29E+03 |
| 1.99E-09 | 95.97706 | 2.09E+03 |
| 1.62E-09 | 94.1903 | 1.89E+03 |
| 1.35E-09 | 92.60668 | 1.72E+03 |
| 1.07E-09 | 90.58768 | 1.53E+03 |
| 8.58E-10 | 88.66975 | 1.37E+03 |
| 6.75E-10 | 86.58608 | 1.22E+03 |
| 5.18E-10 | 84.2866 | 1.06E+03 |
| 3.88E-10 | 81.77663 | 9.20E+02 |
| 2.84E-10 | 79.06637 | 7.87E+02 |
| 2.02E-10 | 76.10703 | 6.63E+02 |
| 1.42E-10 | 73.04577 | 5.56E+02 |
| 9.41E-11 | 69.47179 | 4.52E+02 |
| 6.53E-11 | 66.29826 | 3.76E+02 |
| 4.46E-11 | 62.9867 | 3.11E+02 |
| 2.9E-11 | 59.24796 | 2.51E+02 |
| 1.93E-11 | 55.71115 | 2.04E+02 |
| 1.31E-11 | 52.34543 | 1.68E+02 |
| 8.6E-12 | 48.68997 | 1.36E+02 |
| 5.57E-12 | 44.9171 | 1.09E+02 |
| 3.57E-12 | 41.05336 | 8.76E+01 |
| 2.26E-12 | 37.08217 | 6.96E+01 |

**Tab. 2: Voltage response of the Selinunte sounding and the corresponding transformation into MIDI and frequency.**

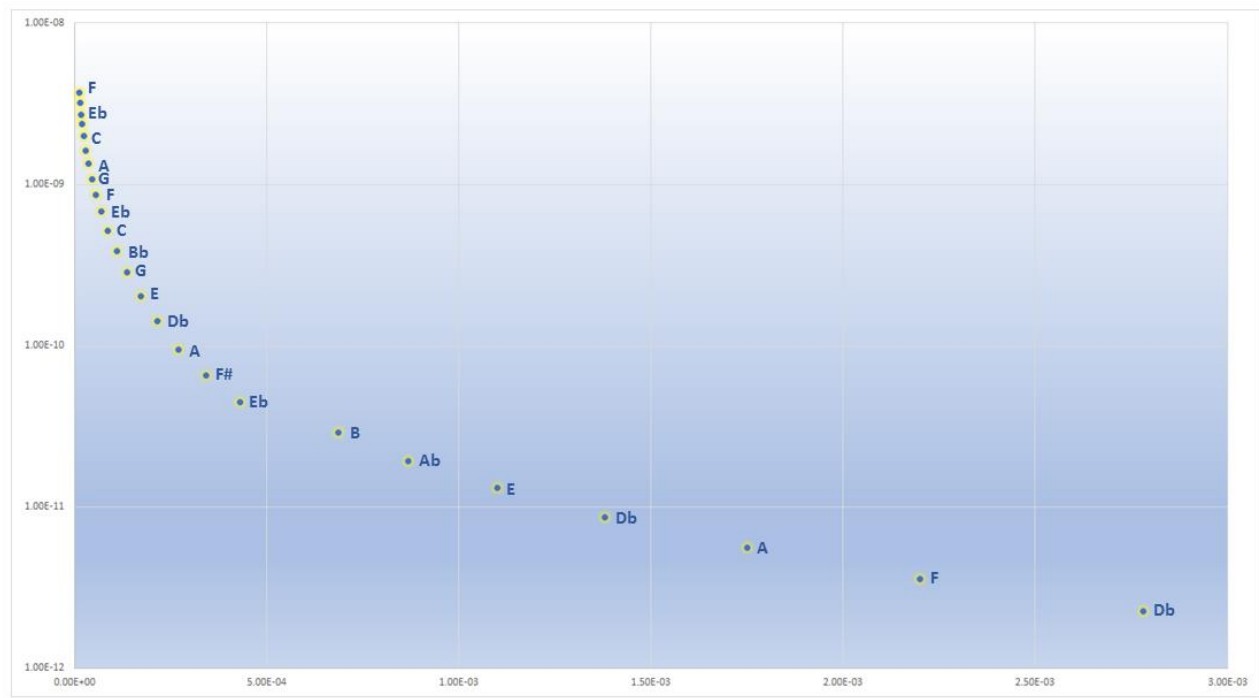

**Figure 3: Sonified transient of a TEM sounding collected over Selinunte temple, Sicily. The Y axis shows the normalized voltage response of the Earth (V/Am$^4$), the X axis the actual acquisition time of the geophysical instrument (in seconds).**


Essentially, we can recognize 4 typical features of a "pure" EMusic composition (soundscape):

1) The pitch is directly proportional to the voltage response: the notes produced by a conductive formation, that produces a stronger response (e.g. clays or shales) are higher than those ones extracted from a resistive rock (e.g. limestones or granites)

2) The pitches must become lower and lower, as the voltage response become weaker and weaker

3) The interval between two near pitches is linked to the resistivity of the material: it will be smaller (chromatism) in the case of conductive formation, due to the slower decay, while it will be wider for resistive rocks, where the eddy currents travel faster

4) The execution time is dictated by the technical specifications of the geophysical instrument, hence the first pitches

185       will be closer and they become ever more distant during the performance.

### 3.3 How EMusic shows are organized

As we state describing some events that already took place (par. 5), an EMusic show preview an active role of geoscientists. During the concert, geoscientists introduce every track for preparing the audience on what they are going to listen. The concert in Ferento that we presented at EGU 2018 (for a video of the event see par. 5.2) can be an example of how the event develops. In that occasion, the audience experienced a journey into the Earth travelling into older and older geological formations. In introducing the first track, a geoscientist explained how the TEM method works and how geophysicists can

model the subsurface, by comparing the decay rate of the transient with the interval among pitches. During the second composition, the musicians began to interplay with the pitches provided by the Earth: we reversed the first track, so that people listened to the return, from the maximum exploration depth (in this case about 100 m) to the surface. The saxophonist and the guitarist were able to improvise over the EMusic base, by using the same pitches, in a sort of natural jam session, where the Earth is the bandleader. Then, we analyzed each geological formation, by showing the musical mood provided

by the relative pitches.

## 4. Engaging with musicians

How the musicians work with the obtained soundscapes? They can choose different modalities. One is to reverse the "pure" EMusic track. Starting from the deeper and late gates/pitches the musicians have the possibility to stay? tuned in with the earth pitches during the return trip to the surface. As the saxophonist Marco Guidolotti played during this reversed part of

Selinunte piece, the relative score has the form shown in Figure 4 (notice that the pitches are translated into an Eb instrument, while the sonification produced notes in the C-key). The musician chose to fix some chords that can be assigned by grouping the pitches so to get some chords.

The whole piece can be listened in YouTube (https://www.youtube.com/watch?v=qsTlMZsGoBE&feature=youtu.be), with the first half composed by using "pure" sonified data, and the second one with the interplay of the saxophonist.

Another modality is to group the pitches according the different geological formations crossed by the EM signal. It requires a modelling of the geophysical data, that can be achieved by the geophysicist. The musicians can use it to compose original pieces or to address the improvisation into more restricted musical scale/chords.

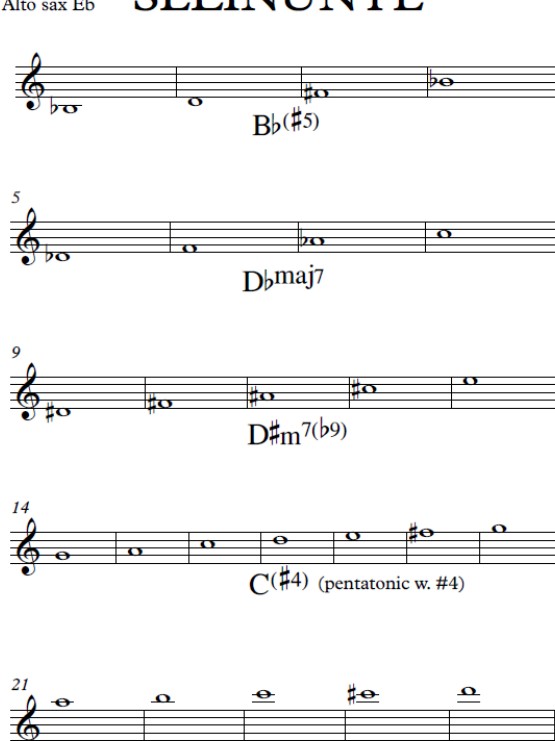

**Figure 4: Score of the Selinunte composition.**

The interplay between the geophysicist, which sonified the data, and the musicians is certainly one of the main step since it represents an effective connection between Music and Geology: besides the pitches extracted from the subsurface, the geophysicist/geologist provides some keywords that can suggest ideas for the improvisation/composition by the musicians. These keywords are associated with the modality of deposition of the formations, paleogeographic info, specific colours associated with the rocks. E.g., for the pitches drawn from a volcanic rock, the keywords could be: hot, fire, red, explosion. Moreover, when the EMusic is performed live, the musicians are inspired directly from the landscape, as in a kind of "audio land art". E.g. for the Vesuvius concert the Quartet that performed the concert were obviously inspired by an excursion to the crater, being captured by the astonishing scenario. Also, the stage location immediately at the foot of the main crater Piazzale Ercolano at 1000 m a.s.l.), contributed to the artistic outcome.

All the musicians involved in the project has been really enthusiastic, having the opportunity to work on an original approach to improvisation/composition. Similar to "standard" pieces played in jazz music, they need to start by a pre-defined framework, formed by notes, chords, modes, while at the same time having the advantage to benefit from a higher degree of freedom: the

geophysicist only provides the pitches and the keywords, without influencing any musical feature (choice of the instruments,
kind of arrangement, development of the track, and so on). Of course, this ability is not specific for jazzists, since it can be easily achieved by any performer of contemporary music.

## 5. EMusic Live events

In the following, we describe the main EMusic shows performed in collaboration with INGV. In each occasion we had the
opportunity to interact with different audiences: scientists, geo-tourists, children and families. From the description of the following events, it is clear that the events may not always follow the same line-up. Sometimes the geology of the place of data acquisition is discussed during the concert, in other cases, as the Vesuvian concert, it can even be introduced before the concert begins.

### 5.1 Performing at the EGU General Assembly

At the EGU Assembly we had the opportunity to perform twice (2017-2018). In 2017 the show took the name of "Sounds from the World" while in 2018 the show was named "Sounds from the Geology of Italy" (Figure 5). The first was based on data collected in Russia (Siberia), Sierra Leone (Nimini), Canada (British Columbia) and Italy (Castelluccio Plain), while the second used data coming from Sicily (Selinunte), Campania (Phlegrean Fields), Umbria (Castelluccio Plain) and Veneto (Venice). A 2018 excerpt is available on https://www.youtube.com/watch?v=qplHWpKPFr4&feature=youtu.be

Since the audience was mainly done by geoscientists, in both cases the show was introduced - in the same format used for a geo-conference - by a formal description of the geological features and evolution of each sites, including seismic and volcanic risk, ice ages, ore-bodies origin (diamonds and gold) and global warming. Unfortunately, due to the amount of work during the General Assembly, the EGU organizers did not effectively promote the two events and the participation was low (meanly about 20-30 people). Anyhow, we received sincere congratulations from those present. The amount of visualizations of the
excerpt on YouTube (231), subsequently achieved confirm a positive feedback.

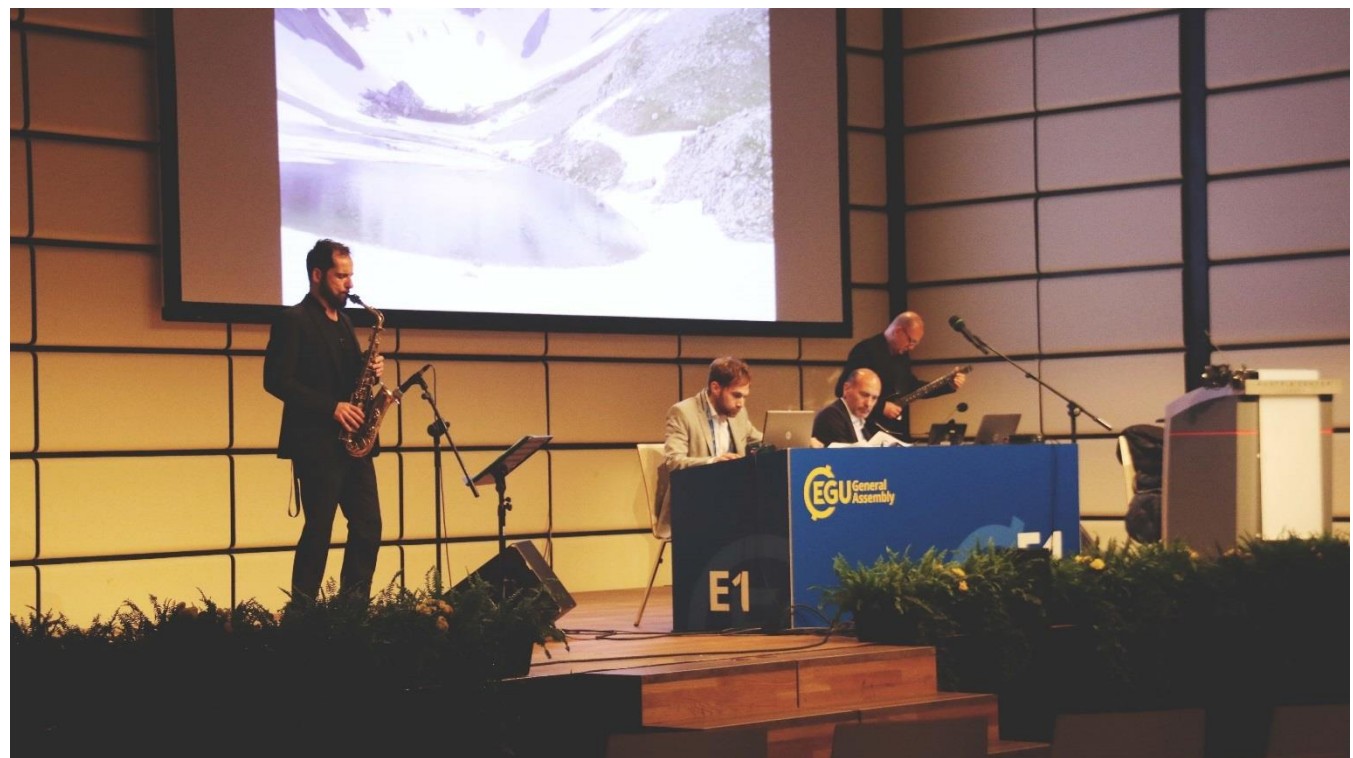

**Figure 5: A moment of the "Sounds from the Geology of Italy" show at EGU2018.**

The EMusic project and its ongoing activity in collaboration with INGV has been presented in several occasion at the EGU General Assembly. The General Assembly is the annual venue of EGU, the greatest in Europe gathering geoscientists from all over the world. The Assembly includes also outreach sessions, and since 2015 also a session on Earth sciences and Art. So, we arranged the presentations in order to capture the attention of a wide public.

In 2017, a poster launched the first pilot project, the MUTENAGE (Musical Tools for Enhancing the Awareness of Global

Emergencies) Project, illustrating five specific global environmental emergencies: pollution of aquifers, seawater intrusion along the coastlines, seismic risk, drought and permafrost melting. Our objective was to make immediately clear how for each of these, the TEM method can be an excellent diagnostic tool, as the voltage response is greatly affected. When we add music, by associating well-defined musical "footprints" to the geophysical variations, through the sonification process, we suppose that the impact of climatic-environmental changes can be perceived also by common people and students of every age and

grade. Of course, this presuppose the spreading of the EMusic method, and the effort of understanding how it works also as a geo-physical diagnostic tool. As an immediate example of how the EMusic can describe the geological features of a site and the environmental issues related to it we chose the effect of seawater intrusion. The phenomenon is well marked by the progressive increase of the voltage and hence of the pitches, when approaching the coastline. When it is "translated" into EMusic, the musical sequence gives an immediate perception of the environmental impact caused by the erosion of the coasts.

But it was only in 2018, that we had the possibility to give an immediate taste of the EMusic to the scientific community when we presented in the Earth sciences and Art PICO session. The PICO is a recently born way of presenting at a scientific conference. Compared to a poster session, a PICO is more suitable for a presentation including Art. It allows you to have a couple of minutes to introduce your work. Then you reach an interactive screen to receive anyone interested and show your work in detail. We had the possibility in our "Diatomites sound like a B13 chord" presentation, to show an excerpt of an EM

concert in the Ancient Roman Theater of Ferento, Central Italy (https://www.youtube.com/watch?v=IaQLhoEQi84&feature=youtu.be).

The session was crowded, and we had several geoscientists already involved in other geo-musical project asking for detailed information at our widescreen. The video on Youtube has reached 458 visualizations, a remarkable outcome considering the scientific feature. As we discuss later, the single tracks of the Ferento concert have been uploaded on Spotify, obtaining positive

feedbacks, with an average of 50 streams.

**5.2 The Vesuvian Concert**

In another occasion we had the opportunity to arrange the performance in a way different from the show in Ferento previously described. It was a show halfway between geo-tourism and a musical concert. On 21st September 2019 we performed at the National Park of Vesuvius in collaboration with the Geological Survey of Campania Visitors assisted the TEM data collecting

(Figure 6), being able to experience the geophysical equipment at work. They asked several questions, and some people were particularly intrigued by the instruments, confirming the advantages of a direct experience.

The EM response was sonified on-site under the eyes of those present and the pitches were immediately transferred to the musicians (the Marco Guidolotti Quartet and the EMusic artistic director Stefano Pontani) to prepare the concert. As the subsurface was formed by very resistive rocks (lava and scoriae of the last activity in 1944) the signal was weak, so that it fell

very quickly into the background noise. Moreover, due to the high resistivity, the pitches were low. We extracted only 6 useful gates and this outcome greatly limited the possibility to arrange a full concert of more than 1 hour. We involved the audience in the contingencies of the case, having also the opportunity to discuss with people the limits of Science. Scientists can get unexpected results, not always positive. Being aware of this risk, we had already thought to use a second TEM sounding collected over a more favourable situation: in fact, this test was carried out in a quarry where we had the opportunity to

characterize the historical pyroclastic flow responsible of the destruction of Pompei, until the older layers of the Somma volcano that preceded the Vesuvius building. Thus, we were able to split the transient into three different pieces. The full concert is available on YouTube, starting at about minute 40

(https://www.youtube.com/watch?v=Xh_tY22E1_A&feature=youtu.be&fbclid=IwAR0bqdEHslm1pD-Up8z_uiMQ7dxvODISckwlDhZOe4slWIwyvMIZ7JGaU_I)

Also in this case, the feedback can be considered promising, with 291 visualizations (11 likes).

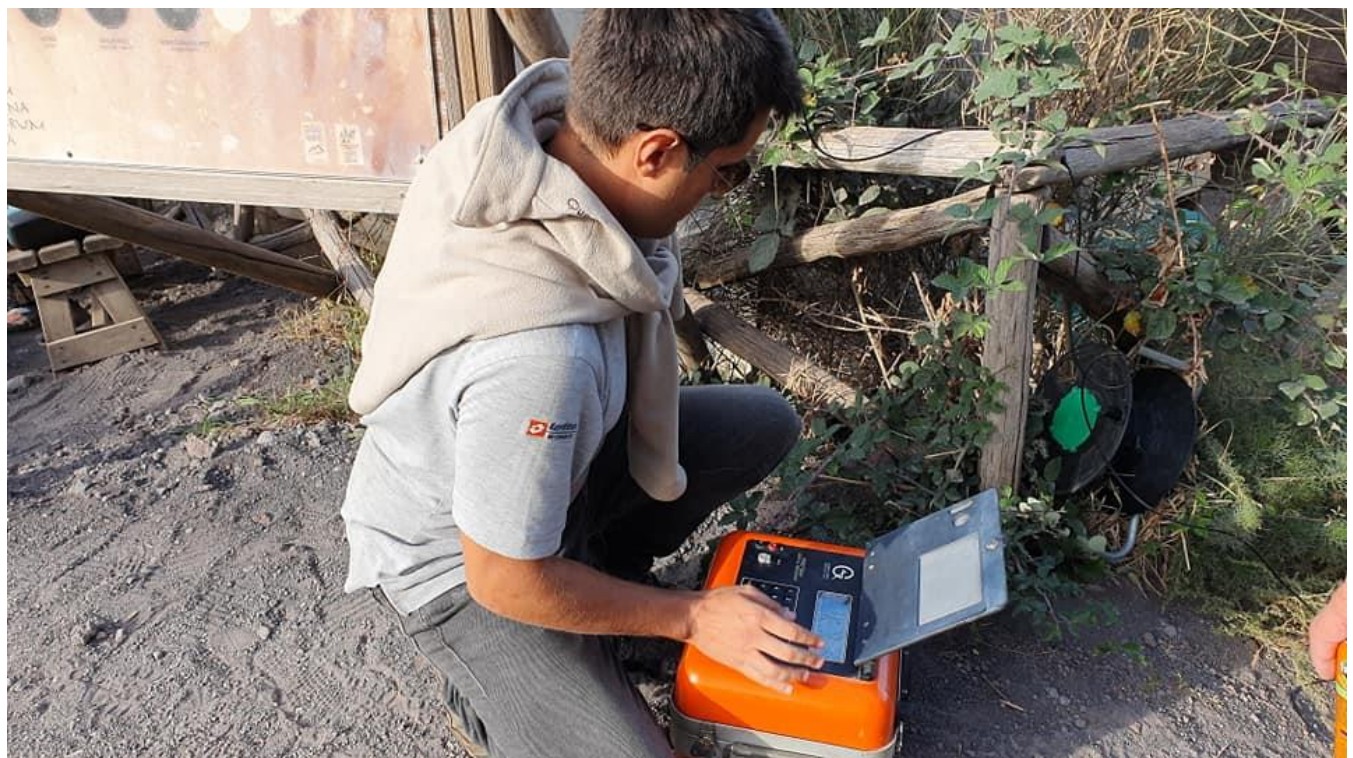

**Figure 6: TEM data acquisition at the top of Vesuvius Volcano.**

### 5.3 INGV Open Day

On 29th September 2019 we were at the INGV Open Day, during the celebration of its last 20 years of research activity and scientific achievements. (Figure 7). It was a rare opportunity to involve families and people of different educational background. We performed three different EMusic pieces inspired by three topics popular in the media: seismic risk, volcanic risk and environmental pollution. The EMusic band, formed by Marco Guidolotti on sax, Stefano Pontani on guitar and Riccardo Marini on electronics, twitted with INGV researchers presenting on the subjects. Musical compositions were based on EM data collected in Castelluccio Plain (close to the Mt. Vettore fault that triggered the last seismic sequence in Central Italy), Australia (in an area affected by seawater intrusion) and Vulsini area, Northern Latium (where the Pleistocene volcanism built huge structures, like the largest volcanic lake in Europe). We combined live music with videos, specially edited for the event and resembling the three main topics of the EMusic compositions. The videos showed images coming from volcanos all around the world as well as from the Australia lands and of the national park of M. Sibillini.

The Open Day involved more than 2000 people and about 300 of them organized in different group of about 30 peoples each spent, at least, 20 minutes in the Conference room listening to our concert. For the appreciation of the whole day we have some

preliminary data as reported from a sample of 150 examined opinion poll survey (89.6 % satisfaction for the whole Open Day event - Crescimbene personal communication). We also had the opportunity to talk directly to the people outside the

Conference room and after the concert, asking them questions like "what did you get from the Emusic?" or " Why did you like or not the music itself?". From the informal interviews to students and families or even to INGV colleagues we perceived enthusiasm. Most interesting was that students mainly catch the change in intensity and duration in time of the music as due to change in subsurface geology while they also appreciate the video and their combination with live music. As from our INGV colleagues, they were satisfied with the way we decided to bring people close to the geosciences.


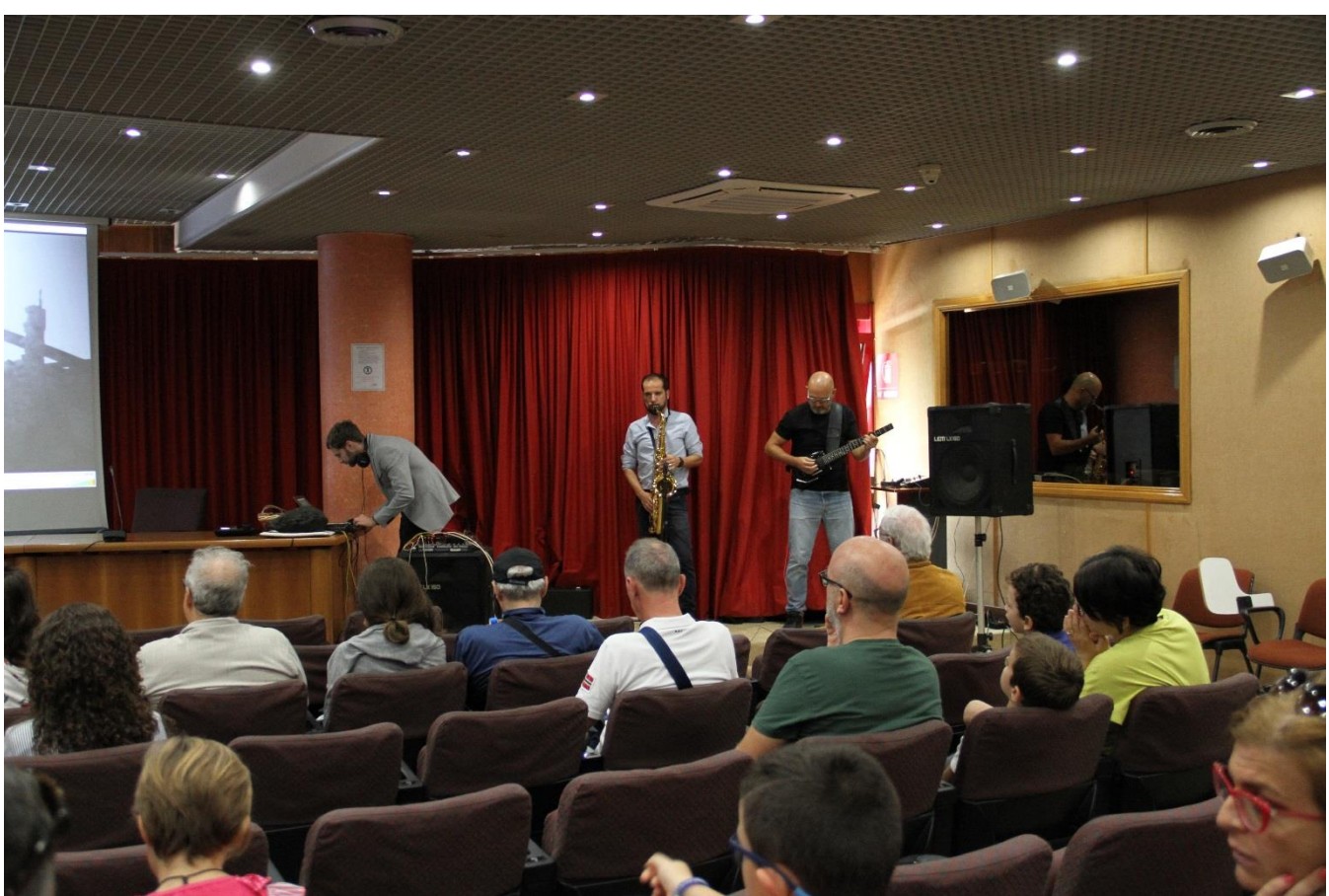

**Figure 7: INGV Open Day in Rome.**

## 6. Discussion

As a project based on sonification, we can already count on a series of studies already performed on what makes this technique so compelling to the public imagination. Sonification has already attracted the interest of scholars in the social sciences and humanities (Stern and Akiyama 2012; Schoon and Volmar 2012; Harenberg and Weissberg, 2012; Rumori 2012). Further studies discuss the experience of sonification in terms of its promise to create sublime experiences of science (Supper, 2014). As far as we stay as a live band performing music created in the interaction between science and art, we can in all respects fit

into what Supper, reporting the thought of Born and Barry (2010) defines the logic of accountability and the logic of ontology. Accountability is referred as art and art-science initiatives used to legitimate scientific research. As Supper observes, reporting several examples of such collaborations in the sonification field, such legitimation is not always a one-way street. Rather than art, making science more accountable to the public, art and science are involved in an act of "legitimacy exchange". In our particular case, who knows if the EMusic will help increase the audience of jazz and improvised music. As for the logic of

ontology, Born and Barry explains that it is referred to "altering existing ways of thinking about the nature of art and science, as well as with transforming the relations between artists and scientists and their objects and publics". In this sense, the EMusic surely suggests new ways of creating music, stressing the role of improvisation, while making the relationship with scientists essential in this area.

## 6.1 A preliminary feedback

Concerning the impact of the EMusic on the public at present we can count on some data extrapolated from our web channels as you tube and Spotify. The data can just give indications on the interest aroused and on the liking. We report them knowing that they are purely indicative having never made an advertising campaign, given that such an action requires a budget. On some occasions we have received media coverage that have contributed to raise the interest of people before and after an event. The Vesuvian Concert was covered by a whole page on one of the most popular Italian newspaper, La Repubblica (Fig. 8), a

video from the same newspaper (https://www.youtube.com/watch?v=WlrcSqxeE0k), scored 1764 visualizations (with 32 likes and 1 unlike). Another article appeared on its scientific supplement. Not least, the event was subject of a question of a popular TV quiz broadcasted by the National broadcast RAI 1 channel.

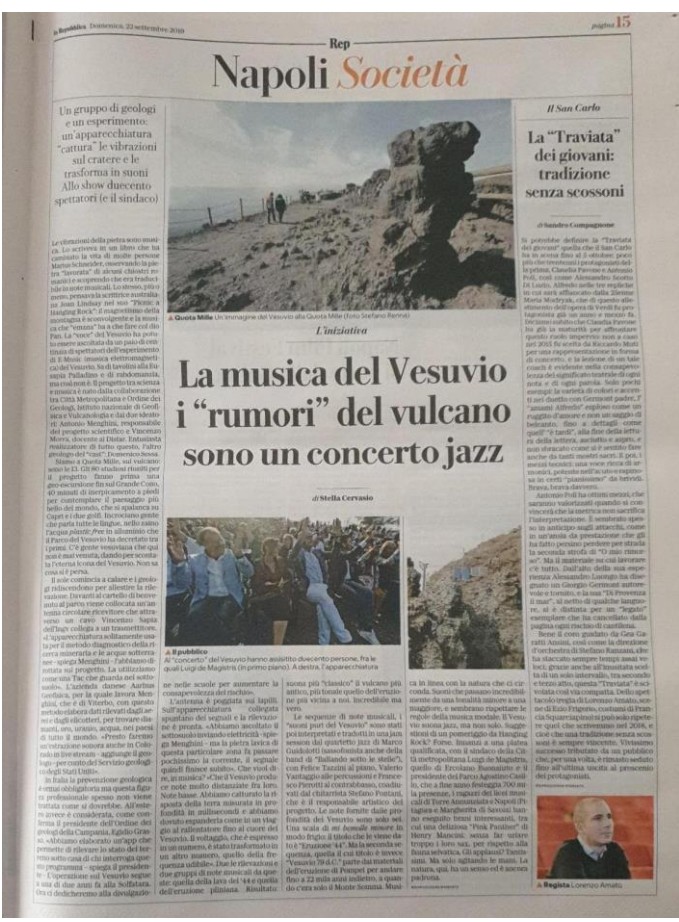

**Figure 8: A full page of La Repubblica newspaper dedicated to the Concerto Vesuviano.**


We tried to perform a statistical analysis by assessing our main channels and media tools. We have two YouTube channels (one under Antonio Menghini: https://www.youtube.com/channel/UCJvHljuFPLHZddyPMghtfzw and another under EMusic Team: https://www.youtube.com/channel/UCLuh1HyrSzIMiAPWGmzPm9g) scoring a total of 32 subscribers. They contain mainly music videos, most of them reporting EMusic events.

In his paper on YouTube videos Allgaier (2012) claims, reporting Watercutter (2011) that if the science in the clips is accurate and valid and if they are entertaining ad gripping too, they have the potential to become helpful tools in science education and science communication. To this respect our activity on you tube has also been addressed to spread the science behind our music. An encouraging feedback in this sense comes from the most viewed of our YT videos: "Listening guide to EMusic" scoring 948 views and an average view duration (AVD) of 1:56 min suggesting the interest to understand more about our

method of sonification from an audience coming not only from Italy (28 %) but also from US (9 %) and Australia (1 %), followed by a large international presence, from at least other 10 countries.

The second most viewed video is "I Suoni della Viterbo Sotterranea" (704 views and AVD= 0:54), connected with a geo-touristic site in Central Italy. In this case very probably the users are visitors of the related geo-site. In that occasion, some people purchased the EMusic cd containing the tracks drawn from this particular site. So far, we sold 200 cd, that can be considered a positive outcome, since the geo-touristic site is the only selling point (besides a small amount sold during the concerts). Also in this case, we appeared in the local press.

The scientific video "Induced Polarization" (624), occupies the third position suggesting the didactic value of the project: the video is not linked in anyway to concerts or events and is most likely viewed by students or researchers. It is an original way to explain the complex phenomenon of Induced Polarization in EM method.

Concerning the video reporting events in collaboration with INGV, the EGU shows "Sounds from the World", one of the tracks, "Sounds from the Fault", scored 586 views and AVD= 1:25 and "Sounds from the Geology of Italy" was viewed 231 times with AVD= 1:45, The Ferento concert presented at the PICO poster has 458 views with AVD= 1:31 and the video associated with the oral presentation "Flying through an African Greenstone Belt" has 360 views and AVD= 1:59.

As we have already highlighted, the feedback of the Vesuvius Concert has been largely positive, with high rate of visualizations of the La Repubblica video (1764 views) and of the full concert (291 views).

The examination of the streams by Spotify provides a partial view of the musical outcome of the project, as the listeners, in this case, do not have any complementary information about the project (listening guide, papers, etc.). We can state that they are more interested to the musical feature. Indeed, we uploaded only 15 EMusic tracks, mostly coming from the Ferento concert. The most streamed track is "Sounds at the foot of an extinct volcano" (212 streams), that is not linked to any show. The second track is "Sound from the Fault" (199) that is in anyway linked to the EGU events, as well as "Halfway River" (173), one of the tracks of "Sounds from the World". The total streams scored 955.

We have a discrete group in Linkedin: https://www.linkedin.com/groups/8509853/ (208 members) and a Facebook Page: https://www.facebook.com/emusic.team/ (with more than 5000 subscribers). As from the Earth&Art blog where a post was dedicated to the Vesuvian Concert on Sept 2019 (the only post in Sept. 2019) the statistics (see Tab. 3) seems to show an increase in the number of visitors, with respect to the previous and subsequent months. The increase could be linked to the media coverage.

| Month | Unique Viewers | Average visitors | Visited pages | Average page views |
|---|---|---|---|---|
| 03/2020 | 974 | 31 | 2.618 | 84 |
| 02/2020 | 971 | 35 | 2.337 | 83 |
| 01/2020 | 920 | 30 | 1.497 | 48 |
| 12/2019 | 1.177 | 38 | 2.426 | 78 |
| 11/2019 | 1.850 | 62 | 6.243 | 208 |
| 10/2019 | 4.024 | 130 | 6.400 | 206 |
| 09/2019 | 3.706 | 124 | 9.895 | 330 |
| 08/2019 | 3.633 | 117 | 8.509 | 274 |
| 07/2019 | 3.477 | 112 | 5.241 | 169 |

**Tab. 3: Statistics of the Earth and Art blog for the September 2019 month when the Vesuvian Concert took place.**

As already said, we don't have a budget to run any promotional activity, as it usually occurs for musical events. EMusic events are financed by sponsors, scientific or professional institutions, without any support from the show business. Usually the budget is used to pay musicians who are professionals coming from the Italian jazz scene.

Nevertheless, if we want to move in the field of science education, an evaluation becomes essential on how the method can help students to get interested in Earth sciences and how the EMusic can sensitize them to planet sustainability.

For the moment we can only suppose it. Eventually, as we have perceived during the musical events organized, EMusic can stimulate people to get interested even on the functioning of sophisticated equipment and the physics beyond a complex method like TEM. At the same time, EMusic can stimulate people's curiosity on how rocks are characterized by different physical parameter (in this case resistivity) and how geoscientists exploit this feature to explore the subsurface.

The presence of musicians can stimulate to study in deep the relationship between frequency and musical notes, the use of the

tempered scale, how a series of pitches can suggest a mood reflecting a specific formation, how the musicians face the improvisation rules.

Since we are convinced of the potentialities of the method from a science communication perspective, we expect to bring soon EMusic in secondary schools. At present we are implementing the "Georisonanze" Project. In this way, we hope to encourage students of scientific institutes to approach Music and students of musical-artistic schools to understand the utility of STEM

subjects. A preliminary description of how we intend to conduct the evaluation is given in the next paragraph.

### 6.2 Future plans

In the near future, we intend to stimulate the involvement of the scientific community in our project and implementing our participation in specific musical events. We are planning the participation to science festivals, not only in Italy but also abroad. There is a growing interest in events attracting people of different ages, young students and families in learning science in an enjoable way. There are already studies on the positive impact of science festivals on the public understanding of science (Jensen and Buckley 2012) We are also interested in participating in musical festivals, those having an experimental and more contemporary feature, with a particular attention to the jazz avant-garde world. In this way we can experiment new perspectives.

We are also currently planning how to get feedback from the heterogeneous audience participating in our show. To start, very probably, a questionnaire will be put on the EMusic website. It will mainly investigate the level of appreciation and the attitude of those present towards science and the environmental issues proposed during the events. Later, to encourage feedback, if we manage to get a budget, we can also think about designing an App for users to keep informed about the EMusic events and download the questionnaires to be filled out.

A more articulate evaluation can be previewed for the project Georisonanze, conceived for the students of secondary schools. In this case, not being a spot event, we can investigate what students retained from the geological and musical knowledge transferred not only through traditional front lessons but also through jazz musical sessions, where the basis will be the sonification process described in the present work. As for previous experiences with schools including other forms of arts (narrative) already carried out at INGV (La Longa et al 2013; Lanza et al. 2014; Lanza e D'Addezio 2020) an input questionnaire will investigate the students' previous knowledge on geosciences and music, while a final questionnaire will assess the level of appreciation, the acquired skills and other items still to be discussed. While, concerning the evocative power of music, it will be interesting at the end of the didactic modules to propose an oral quiz session all together to see if the students through the sounds proposed by the musicians are able to recognize the rocks and the nature of the soil.

### 7. Conclusions

Until now the EMusic have been experimented under the form of live shows worldwide. When the audience were composed by scientists, students, geo-tourist, children and families, we perceived its potentialities of attracting people interest on science. A preliminary feedback received from the EMusic presence on the web encourage us to assess the EMusic as a science communication tool to get people in contact with the geology of the Earth, spanning from natural risks (seismic, volcanic and geomorphological) to climatic changes, from pollution issues to landscape evolution. Music is certainly a unique medium to raise awareness of the most urgent topics that threaten Earth. The peculiarity of the TEM methods at the basis of our sonification process, has given us the opportunity to taste the predisposition of people to get interested in the investigative methods of earth sciences even the most sophisticated ones. The first positive feedback in this sense came from the musicians. But on other occasions as, for example, the Vesuvian Concert, we have also found that ordinary people can be intrigued by

scientific equipment and their use. Our next step will be bringing the project in schools, since we are convinced that coupling music with geosciences can emphasize the central role of STEM matters in our society encouraging people to appreciate the use of Maths and Physics in an artistic context.

**Acnowledgments**

We wish thank Domenico Sessa (Association of Geologists of Campania) for having supported the Vesuvius Concert, Dora Apicella for the photo of the Vesuvius Concert, and Massimo Crescimbene (INGV) for the invitation to the INGV Open Day.

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
