# Peer review of "ElectroMagnetic Music: a new tool for attracting people interest in Geosciences, while sensitizing them to planet sustainability"

_Geoscience Communication, 2020_

## Referee Comment (RC1) · Anonymous Referee #1 · 23 Apr 2020

The paper concerns a project aimed at converting Transient ElectroMagnetic Method (TEM) data, which provides information about the resistivity of rocks, into music. This is a topic which will be of interest to Geoscience Communication audiences. There is real potential in this manuscript to describe the process of how these sonifications and collaborations with musicians and/or composers occurred, however, at present the manuscript is incredibly confusing and lacking crucial details. In the interest of collegiality I raise a number of concerns which I hope the authors can address to improve what could ultimately be an interesting contribution to this emerging field.

The introduction does not adequately frame this work in the either the context of the
wealth of sonification projects or in the field of TEM analysis. At present this section reads like an extended abstract, which is confusing to readers. While the authors do provide examples of datasets which have gone through sonifications, nowhere is a definition of sonification in general given neither is the diverse number of approaches to sonification (such as direct audification, mapping datapoints to MIDI instruments based on values, model- vs data-based sonification etc.) discussed. These are crucial aspects required in order to better understand the work presented.

Section 2 enables readers to better understand what the data behind this project actually represents. However, many questions still arise. Figure 3 is presented in the text as being standard TEM data, but appears in the figure and caption itself as a sonification. Could the authors provide the TEM data in the raw format that would be the usual for a scientific publication within the field, perhaps labelling so that those from other fields can understand it. Merging this figure with the sonification so early in the manuscript merely confuses the readers and perhaps two separate figures are required, or at least two different panels?

The process of the sonification itself is not adequately described. A flow chart would be very helpful in this regard along with technical details, any choices made by the authors and how these were justified. For example, what choices of maximum and minimum voltages were made and how were these determined? What are the limitations with these choices given known ranges of TEM events in the literature? And what audible frequencies / notes were these voltages mapped to? Were considerations made based on different musical instruments?

Interaction with musicians and composers seems like it deserves a section of its own. How were these collaborations established, what were their initial thoughts to the sonified TEM transients? Were they involved in the design of the sonification or only approached afterwards? These are important considerations within the scope of the journal.
The sections devoted to events also lack a lot of required information. What the format/sessions within the EGU General Assembly are not currently clear, and for readers unfamiliar with this conference a discussion of the types of attendees and presentation methods is required (e.g. a PICO is not described). Further, how the different sessions operated is not explained. What were the purposes of the different sessions? Were the musical performances used to explain, or did they simply follow the explanations? Similar considerations are required with the other events described, where it is not clear who these sessions were aimed at (scientists, geoscience-interested publics, non-science audiences?), how they were targeted, and what different considerations to the presentations were made to be appropriate to these different audiences.

It is clear that no formal evaluation of the activities has been performed, which is a real shame. While the authors claim audiences "greatly appreciated" the events, without even a description of how professional observations were made these claims are unfounded. Instead I would urge the authors to be more honest and instead present their "feedback" section more as a critical reflection of their own practice and as future plans for this project, detailing how through evaluation they aim to understand the impact these sonifications may have on different audiences, because at present no claims can be made about this.

Similarly, the conclusions should not over-claim what has occurred and been shown to be effective in science communication or engagement. The sonification process of TEM data and how the authors have engaged with musicians would be suitable enough for publication in Geoscience Communication, whereas the impact on attendees and media is too weak at present to offer any firm conclusions.
**Discussion** paper

---

## Referee Comment (RC2) · Bernardo Feldman (Referee) · 8 Jun 2020

I will begin by acknowledging that my background is in music with a profound, yet tangential interest in science. I read the manuscript and I watched and listened to the included videos of the various events as they were included within the paper.

Thus, as I went through the process, I made notes based on my own questions and as the manuscript and the music triggered my own curiosity.

I am including portions of the particular paragraphs from where I extract my thoughts. It is these notes that I am forwarding to you. Please use them to the degree that they

might illuminate your own perspectives on the subject. Since the format of the does not allow for different fonts or any other way to assist the reader in differentiating the manuscript's text from my own comments, I did place quote marks around the extracted words from the original paper.

"Since the beginning, a band of musicians experimented the EMusic giving concerts all over the world covering different geological locations. The sound representing each scenario was recovered in situ with the above-mentioned methodology." No explanation YET as to how exactly the sonification took place. Did the musicians translate the data by providing their own personal interpretation? Were voltages used to assign amplitudes (loudness/dynamics), frequencies (pitch values), lengths of individual sounds to create rhythmic patterns? There is no clarification given to the reader who might not know what sonification exactly is. "The audience can experience a journey into the Earth by riding the eddy currents produced by the EM field. Not only it is a travel in space, but also in time, as we explore through EMusic older and older geological formations..." That is a beautiful thing. It needs to be recognized and further emphasized that this represents an effort from both, scientists and musicians, to engage the intellect and emotions from the intended audience, and that the combination of sonified transformation of the aforementioned voltages, along with the involvement of musicians and sound artists, will be what will trigger their curiosity and emotional response. The sonification alone will only awake the interest of a relatively small crowd. Only to the type of listener for whom the mostly static quality of sounds is attractive and understood at an emotional level will be able to keep their attention for a sustained period of time. To them the hypnotic, meditative, and trance-like quality as the sounds progresses from one moment in time to the next will certainly be attractive. Interestingly, traditional ragas from India do possess these qualities. Traditionally, far East cultures do have a different sense of time and a different philosophy about life. Perhaps, at present, and with the revival in the West reading the emphasis toward our learning to live in the here and the now" and through the explosion of the "mindfulness" movement the ability to listen and assimilate static sounds will become more widespread.

**GCD**
"The source of the sonification (a voltage) depends upon the electrical behavior of the rocks (i.e. the resistivity). Following this assumption, the basic principles were codified by Menghini and Pontani (2016) Thus, it is not out of place to claim that we can extract the effective "Sound of the Earth". Similar to other sonification process, we adopt a mathematical rule that allows us to translate the geophysical data into audible frequencies."

Translations based on someone's codifications (such as the ones by Menghini and Pontani) are, by necessity, biased and will be interpreted in radically different ways. Thus, there has to be a recognition that different musicians will arrive at different interpretations of such data. A musician or group of musicians whose musical training came from having studied at a traditional Western institution, (such as at a music conservatory) will most likely be very different from someone who studied jazz, pop or rock or whose background comes from the "popular" musical traditions of parts of Latin America, India, Africa or elsewhere.

"It would be therefore more correct to say that we can produce pitches, rather than sounds." At this point in the paper, the reader doesn't know if these pitches are based on natural harmonics, of based on the Well-tempered system, microtones or some other tuning system. Clearly, any of these choices/translations/interpreatations will produce different results.

"The musical notes can be played by any kind of instrument, also by a human voice." Clearly, and by no means wrong, this is an arbitrary choice of timbre. If the intention of the project is to bring awareness to our planet, it really doesn't matter how literal the translation of the data provided by the voltages is. Evidently, as the title of the article clearly states, it is intention to attract people's interest to geoscience while, in the process sensitizing us to planet sustainability. If, in the other hand, the sonification can to be used as a tool to "prevent risks" and assist in the reading of a particular phenomenon occurring at a certain part of our planet, there has to be a consistent array of sounds or textures that could quickly and easily be understood by people, Interactive comment

scientists in particular. I would equate these as type of "sonic fingerprints" mentioned in line 145 of the article. I presume that there already many other tools are in place to measure potential earthquakes, eruptions, pollution of aquifers, seawater intrusion along the coastlines, seismic risk, drought and permafrost melting, etc.

The involvement of the musicians is direct, as they have to arrange an improvisation or a composition, by using these pitches provided by the Earth: the EMusic data are the bricks that will be used to build any musical performance (some examples have been reported by Menghini, 2016, 2018 and Duncombe, 2019). This allows to achieve an effective connection between Art (Music) and Science (Geology), in a way that can be easily understood by common people. "How easily understood by "common people" will depend as to how the musicians are translating the data. During the 20th century, various composers have made numerous translations of natural phenomenon. A few of them, not mainstream, and "easily understood by common people" are listed below. Some other examples are cited in the Introduction of this paper just below. Chares Dodge (Earth Magnetic Fields) https://www.youtube.com/watch?v=j5MHsnc67yw

**Larry Austin (Canadian Coastlines) https://www.youtube.com/watch?v=2zs5rEbXbmU**

Mickey Hart (from the musical group the Grateful Dead) Sounds of the Universe. Also, of Brain and heart impulses interaction between music and medicine https://www.pbs.org/newshour/show/big-bang-cosmic-vibrations-grateful-deads-mickey-hart-plays-rhythm-universe Mickey Hart (from the iconic musical group the Grateful Dead) Sounds of the Universe claims to have captured the vibrations of the cosmos and has invested time in sonified these vibrations while also over imposing his own musical interpretations clearly influenced by his Rock & Roll back-ground. Mr. Hart has also done work with Dr. Deepack Srivastave from Gladstone Institutes at the University of California in San Francisco where a team of stem cell researchers has been working on identifying impulses generated by brain and heart cells to convert the electrical energy generated into sound to be able to map difference between deceased cells and healthy ones. https://www.pbs.org/newshour/show/big-
bang-cosmic-vibrations-grateful-deads-mickey-hart-plays-rhythm-universe

Particularly interesting are the electro-acoustic works of lannis Xenakis, composed utilizing environmental sounds a few of them are listed below" lannis Xenakis (Di-amorphoses) https://www.youtube.com/watch?v=b7235DNgkd0 lannis Xenakis (Con-cret PH) https://www.youtube.com/watch?v=S9zMalhuMgo lannis Xenakis (Orient-Occident) https://www.youtube.com/watch?v=PzVoYt78iZ0

"Finally, Geoscientists can be considered in all respect as composers, or rather as the medium between the Earth and the performers..." Nice!!

"...also by providing some keywords" (keywords? -not quite clear to me...) "on the history of the geological site inspiring the musicians."

We are also implementing a project, named Georisonanze, to bring the EMusic in secondary schools to introduce students to the investigation of Earth with modern techniques while sensitizing them to planet sustainability." (I assume that this is an adaptation, or an extension or the sharing of the experiments mentioned above. Am I right?) "This paper describes in detail the method of sonification..." Never in the manuscript do I get a detailed description of the sonification method that is clear to me.

"We describe also the potentialities of the methods from a science communication perspective, even though we never conducted a survey to test the efficaciousness of the methods."

What exactly do they mean by testing the efficaciousness of the method? "Nevertheless, we preview in a near future to experiment the method in schools to have the opportunity to conduct a survey for investigating the efficaciousness of the method (we are implementing a project named Georisonanze)" What exactly are they looking to find through these experiments in schools? Are they looking to measure how students respond and get sensitized in regards to planet sustainability? Is it by simply bringing attention to the subject? Early in the manuscript, a clear recognition needs to be GCD
made to the fact that a marriage of sonified interpretations of the voltages generated by earth's subsurface materials does take place with the involvement of performing musicians that embed these data with an emotional element that thus gets enhanced. As I listened to the shared video (shown right below) the points that I made above made it clearer that the paper itself.

https://www.youtube.com/watch?v=qsTIMZsGoBE&feature=youtu.be

"As the transient works out within a few milliseconds, we adopted a time expansion that can be chosen depending on how long we want to arrange the composition (usually we use values between 100,000 and 1 million). Otherwise our ear would hear a single chord formed by all the gates/pitches."

From a musical point of view, the formation of chords, which is precisely the manifestation of a collection of pitches occurring simultaneously, represents a welcome element within the musical discourse. Their activation through arpeggiations and other means to create accompaniments as the middle ground of a composition is something that composers and performers pay close attention. The expansion over time as explained above makes sense. Their distribution over longer periods of time, as explained in this paper and demonstrated throughout the link to the video allows for easy analysis of their frequencies making it easy for musicians use this information to create melodies that are coherent with these collections.

"In order to make comparable sonified transients, we prefer to use the same range of voltage, i.e. fixed values for minimum and maximum response: this device allows to compare different geological scenario and different EM systems in an objective way. This approach can be similar to the choice of the edges of a colour scale to figure results." Although it is stated that fixed values are used to assign to the various transient's particular pitches in an objective manner, it is not quite clear how these pitches are arrived at. The analogy comparing it with edges of a color scale does help to assume that gradations have been arrived with a certain logic, most likely translating

GCD
numerical values into audible frequencies that retain precise intervallic relationships. The explanations right below help in confirm that amplitudes (louder sounds) will come from the stronger responses and vice versa, and that higher frequencies (pitches) will correspond to stronger responses. Inasmuch as I tried to find a clear correspondence between the pitches given in Figure 3, arrived at as transformations of the voltage's responses, and the so-called "score" utilized by the saxophonist to interplay with these collected sonified transients. Although there is no explanation as to how the player choose these particular chords as the basis for his improvisation, while listening to the two elements interacting with each other, I found that the performer did utilize pitches that beautifully harmonized and complemented the clusters created within the sonified textures. While crossfading, alternating pitches within the sonification produced a sense of chromatic movement extending beyond the traditional harmonizations found in "traditional" Western music, and closer to the textures of musicians experimenting with pure intonation and microtonalism.

"Another modality is to group the pitches according the different layers crossed by the EM signal. The musicians can use it to compose original pieces or to address the improvisation into more restricted musical scale/chords." This sentence, finally, further emphasizes the degree of involvement, and the limits set to the players as they choose the pitch- sources for their improvisations. It helps the reader to understand how it all can work in various modalities.

"During EGU2018 Assembly we presented a PICO poster ("Diatomites sound like a B13 chord"), showing an excerpt of an EM concert in the Ancient Roman Theater of Ferento, Central Italy 155 https://www.youtube.com/watch?v=IaQLhoEQi84&feature=youtu.be"

Upon listening to this second example it became even clearer that the musician's choice of notes for their improvisations took into consideration the intervallic relationships present in the sonification, yet allowed themselves the freedom to expand from such collections.
"As the subsurface was formed by very resistive rocks (lava and scoriae of the last activity in 1944) the signal was weak, so that it fell very quickly into the background noise. Moreover, due to the high resistivity, the pitches were low. We extracted only 6 useful gates and this outcome greatly limited the possibility to arrange a full concert of more than 1 hour. We involved the audience in the contingencies of the case, having also the opportunity to discuss with people the limits of Science. Scientists can get unexpected results, not always positive. Being aware of this risk, we had already thought to use a second TEM sounding collected over a more favourable situation: in fact, this test was carried out in a guarry where we had the opportunity to characterize the historical pyroclastic flow responsible of the destruction of Pompei, until the older layers of the Somma volcano that preceded the Vesuvius building. Thus, we were able to split the transient into 3 different pieces. The full concert is available on YouTube, starting at about minute 40." https://www.youtube.com/watch?v=Xh tY22E1 A&feature=youtu.be&fbclid=lwAR0bgdEHslm1pD-Up8z uiMQ7dxvODISckwIDhZOe4sIWIwyvMIZ7JGaU I Evidently, there is an effort to bring the public into the experience. Aware of complex circumstances, scientific team and the musicians make choices to make a concert practical and enjoyable. The participation of Jazz musicians is a good choice, since many people relate to their musical vocabulary. As explained by one of the organizers, the improvisation for this event is based on the Eb Phrygian mode. The explanation given at about the 42minute explanation by one of the organizers and with the pianist playing the notes that correspond to the sonification help in making connections much easier than by reading the essay. As the camera moves through the stage, we see the reference notes in the music stand of the sax player. And as we hear the music, we guickly realize that the sonification are but a departure point for all 5 musicians(double bass, piano/synthesizer, percussion, saxophone and guitar) to improvise. Notes other than the ones transformed from the readings of the voltages are added as neighboring tones, suspensions, to enrich and embellish the texture. As I was listening to the excerpts. I looked for the notes in my piano at home and confirmed that the

**GCD**
sonifications traveled through various chords creating very interesting progressions, particularly when combined with the notes added by the musicians. "Since we have never conducted a survey, for the moment we can only suppose that the EMusic have great potentialities for raising interest of people on Earth sciences studies, while sensitizing them to planet sustainability. Our supposition is based on the people's mood we perceived during our several experiences around the world. Eventually, EMusic can stimulate people to get interested even on the functioning of sophisticated equipment and the physics beyond a complex method like TEM. At the same time, EMusic can stimulate people's curiosity on how rocks are characterized by different physical parameter (in this case resistivity) and how geoscientists exploit this feature to explore the subsurface. The presence of musicians can stimulate people to study in deep the relationship between frequency and musical notes, the use of the tempered scale (in theory we could assign to any pitches the effective frequency, i.e. microtones, and not the closest one listed in the twelve-tone scale commonly used in Western Music, since Bach's time), how a series of pitches can suggest a mood reflecting a specific formation, how the musicians face the improvisation rules."

It only becomes clear in the area of the document labeled "Feedback" when the authors expose the fact that the pitches chosen are arbitrarily, and with no direct relation to the readings of the voltages read by their instruments, based on the well-tempered scale commonly used. As the authors point out, it will be interesting to explore the reactions of the people should they choose to assign microtonal intervallic relationships, or based on just intonation, ancient Arab, meantone, Pythagorean, etc. The overtones that I heard I listened to the sonifications do include already intervals smaller than the half step, thus creating fascinating chords, textures and timbres. I venture to say that , because of the places where these events take place, and because of the experimental quality of the context, audience will be more open to listen to the music being created, so long as it doesn't veer to much away from what they are used to listen to. Active performance participation by students through the Georisonanze project will also open them to both, accept music that is outside of their current comfort zone and to develop
the type of sensibility toward our planet, which is, of course, the principal goal of this project.

---

## Author Comment (AC1) · 5 Jul 2020

Thank you Referee#1 for the interesting and necessary observations. Following your suggestions we have extensively reorganized the manuscript starting from the abstract:

The revised abstract:

In recent years the different methods used to translate data into sound to help scientists to better organise their work have come out of the scientific realm to cross into other areas and achieve purposes other than those pursued strictly by scientific research. The ElectroMagnetic Music, a project born in Italy, fits fully into this area. By transform-

ing into musical pitches the voltage response collected by Transient ElectroMagnetic Method (TEM), a well-known geophysical tool for exploring the subsurface, this novel approach enables to extract musical pieces reflecting the effective geological setting, in a way that any geological site seems to have its own soundtrack (i.e. the "soundscape," the audio component of a landscape). The soundscape becomes the basis on which a dedicated band improvises jazz music. Besides being a new method for creating music, our project has the ambitious goal to attract people's interest on Earth sciences and their investigative methods, while raising awareness of the environmental problems that characterize geological sites. The present work refers on the experiences already done by the EMusic as a live band around the world. Reports some preliminary data on people reaction and anticipate some future plans for better assessing the potential of the method as a good science communication tool.

As for your suggestions:

1 - The introduction does not adequately frame this work in the either the context of the wealth of sonification projects or in the field of TEM analysis. At present this section reads like an extended abstract, which is confusing to readers. While the authors do provide examples of datasets which have gone through sonifications, nowhere is a definition of sonification in general given neither is the diverse number of approaches to sonification (such as direct audification, mapping datapoints to MIDI instruments based on values, model- vs data-based sonification etc.) discussed. These are crucial aspects required in order to better understand the work presented.

We have rewritten the introduction following your advices, concentrating mainly on reviewing some research on sonification and audification. We have underlined in what we believe EMusic is original compared to the other projects based on sonification.

We write: "The EMusic (ElectroMagnetic Music) in this trend is certainly the first one utilizing the EM response of the Earth, being the first case of sonification strictly related with the geological structure of the subsurface"

Further: "What distinguishes our project is a real dedicated band that improvises jazz on a musical basis obtained through the sonification process".

We also added a paragraph specifically dedicated to our objectives:

2 Objectives

The EMusic has been conceived to translate into "music" data acquired by a specific scientific instrument, normally used for many geoscience applications. We believe that this technique has a great potential in terms of science and art communication capability. To get a first taste of these potentials, in a first phase, our agenda included mainly live events in several geo-sites. We performed all around the world in close cooperation with musicians to promote the EMusic. We also used the net to spread our method of sonification, the events performed, and the ones scheduled. In the near future we intend to bring the project in schools to involve students in Earth sciences, planet sustainability while introducing them to a different approach to music. For the time being, as a live band, we are satisfied since the project obtained great interest by the scientific and musical communities. The EGU General Assembly in Wien invited us to play twice (2017 and 2018). Geoscience Australia invited us to play in Canberra and Perth; AGU Centennial Grant awarded us with a 5 hours sound installation based on Airborne EM data collected in Colorado Mountains; Under the patronage of the City of Naples, the Geological Survey of the Campania District and in collaboration with the National Park of Vesuvius, we played on the top of Vesuvius Volcano; We also performed at the INGV (Istituto Nazionale di Geofisica e Vulcanologia) Open Day; we carried out a tour of 7 stages "Sounds from the Geology of Italy", based on the sonification of EM data collected in some of the most beautiful natural and cultural sites, involving famous international jazzists, like Enrico Rava and Francesco Cafiso. This paper describes in details our method of sonification and refers on the events performed in collaboration with INGV. We describe also the potentialities of the methods from a science communication perspective even if so far we didn't conduct an evaluation. Nevertheless, as previously said, we preview in a near future to experiment the method in schools to

have the opportunity to extensively evaluate its efficaciousness in terms of attracting students' interest in geosciences while sensitizing them to planet sustainability. To this aim we are at present implementing a project named Georisonanze.

2 - Section 2 enables readers to better understand what the data behind this project actu- ally represents. However, many questions still arise. Figure 3 is presented in the text as being standard TEM data, but appears in the figure and caption itself as a sonification. Could the authors provide the TEM data in the raw format that would be the usual for a scientific publication within the field, perhaps labelling so that those from other fields can understand it. Merging this figure with the sonification so early in the manuscript merely confuses the readers and perhaps two separate figures are required, or at least two different panels? The process of the sonification itself is not adequately described. A flow chart would be very helpful in this regard along with technical details, any choices made by the authors and how these were justified. For example, what choices of maximum and minimum voltages were made and how were these determined? What are the limitations with these choices given known ranges of TEM events in the literature? And what audible frequencies / notes were these voltages mapped to? Were considerations made based on different musical instruments?

Following your instructions we have renamed the par. "Data sonification" in "Methods" and we have subdivided the paragraph into three sub-paragraph:

3.1 TEM methods

3.2 Data sonification

3.3 How EMusic show are organized.

Regarding Fig.3 is indeed a "geophysical" representation of the data, i.e. voltage values vs acquisition time. We have specified in the caption what are the axis unitss. Close to the "raw data" we inserted the pitches deriving from the sonification of each acquisition gate of the instrument. In "Data sonification" we have provided details about

the procedure and we tried to reply to all the comments by the reviewer (e.g. definition of min/max values, audible frequencies, etc.). As we have pointed out, we extract pitches that can be played by any musical instruments: indeed the "Sound of the earth" is described by the different pitches, the intervals among them (i.e. scales, modes) rather than absolute musical notes.

3- Interaction with musicians and composers seems like it deserves a section of its own. How were these collaborations established, what were their initial thoughts to the sonified TEM transients? Were they involved in the design of the sonification or only approached afterwards? These are important considerations within the scope of the journal.

Some of the claims have been addressed in the Method Section. A paragraph that particularly discusses the interaction with musicians have been inserted:

4. Engaging with musicians

4 - The sections devoted to events also lack a lot of required information. What the format/sessions within the EGU General Assembly are not currently clear, and for readers unfamiliar with this conference a discussion of the types of attendees and presentation methods is required (e.g. a PICO is not described). Further, how the different sessions operated is not explained. What were the purposes of the different sessions? Were the musical performances used to explain, or did they simply follow the explanations? Similar considerations are required with the other events described, where it is not clear who these sessions were aimed at (scientists, geoscience-interested publics, non-science audiences?), how they were targeted, and what different considerations to the presentations were made to be appropriate to these different audiences.

A dedicated paragraph to the events (5. EMusic Live events) has been added with the following brief introduction:

In the following, we describe the main EMusic shows performed in collaboration with

INGV. In each occasion we had the opportunity to interact with different audiences: scientists, geo-tourists, children and families. From the description of the following events, it is clear that the events may not always follow the same line-up. Sometimes the geology of the place of data acquisition is discussed during the concert, in other cases, as the Vesuvian concert, it can even be introduced before the concert begins.

The paragraph has been subdivided into three subparagraph describing The EGU events, The Vesuvian Concert and the INGV Open day.

To meet the reviewer's requests, in the EGU events sub-paragraph we have added:

"The General Assembly is the annual venue of EGU, the greatest in Europe gathering geoscientists from all over the world. The Assembly includes also outreach sessions, and since 2015 also a session on Earth sciences and Art. So, we arranged the presentations in order to capture the attention of a wide public."

And also:

"But it was only in 2018, that we had the possibility to give an immediate taste of the EMusic to the scientific community when we presented in the Earth sciences and Art PICO session. The PICO is a recently born way of presenting at a scientific conference. Compared to a poster session, a PICO is more suitable for a presentation including Art. It allows you to have a couple of minutes to introduce your work. Then you reach an interactive screen to receive anyone interested and show your work in detail." For the Vesuvian Concert (par. 5.2) details have been added on the different arrangement of the event:

The sentence "In another occasion we had the opportunity to arrange the performance in a way different from the show in Ferento previously described. It was a show halfway between geo-tourism and a musical concert." Introduce the paragraph stressing the fact that in this occasion a part of the public was constituted by geo-tourists

While in the Open Day (5.3) details have been added on the appreciation of the whole

day and on the informal interaction with the public of the concert, in this case formed by students, families and researchers.

5 - It is clear that no formal evaluation of the activities has been performed, which is a real shame. While the authors claim audiences "greatly appreciated" the events, without even a description of how professional observations were made these claims are unfounded. Instead I would urge the authors to be more honest and instead present their "feedback" section more as a critical reflection of their own practice and as future plans for this project, detailing how through evaluation they aim to understand the impact these sonifications may have on different audiences, because at present no claims can be made about this.

To meet the reviewer's requests, at the place of the feedback paragraph a more articulate session now appears.

6. Discussion

Here a hint to some studies on the field of sonfication from the perspective of social sciences has been added. In particular, a study by Supper (2014) of interest fir our particular case:

"As a project based on sonification, we can already count on a series of studies already performed on what makes this technique so compelling to the public imagination. Sonification has already attracted the interest of scholars in the social sciences and humanities (Stern and Akiyama 2012; Schoon and Volmar 2012; Harenberg and and Weissberg, 2012; Rumori 2012). Further studies discuss the experience of sonification in terms of its promise to create sublime experiences of science (Supper, 2014). As far as we stay as a live band performing music created in the interaction between science and art, we can in all respects fit into what Supper, reporting the thought of Born and Barry (2010) defines the logic of accountability and the logic of ontology. Accountability is referred as art and art-science initiatives used to legitimate scientific research. As Supper observes, reporting several examples of such collaborations in

the sonification field, such legitimation is not always a one-way street. Rather than art making science more accountable to the public, art and science are involved in an act of "legitimacy exchange". In our particular case, who knows if the EMusic will help increase the audience of jazz and improvised music. As for the logic of 375 ontology, Born and Barry explains that it is referred to "altering existing ways of thinking about the nature of art and science, as well as with transforming the relations between artists and scientists and their objects and publics". In this sense, the EMusic surely suggests new ways of creating music, stressing the role of improvisation, while making the relationship with scientists essential in this area.

Par. 6 is then subdivided in other two sub paragraph:

6.1. A preliminary feedback

6.2 Future plans

In the preliminary feedback we have extrapolated data from our web channels and social media (EMusic is present in you tube, Spotify, facebook and linkedin,) The analysis was intersected in some cases with the media coverage.

"Concerning the impact of the EMusic on the public at present we can count on some data extrapolated from our web channels as you tube and Spotify. The data can just give indications on the interest aroused and on the liking. We report them knowing that they are purely indicative having never made an advertising campaign, given that such an action requires a budget. On some occasions we received media coverage that contributed to raise the interest of people before and after an event..."

In the par. "Future plans"

We have outlined some ideas on how to conduct future evaluation non only in the case of the heterogeneous audience participating in the show. A project for school, Georisonanze, is on its way, and the evaluation will be conducted following the experience already done for other school projects including art already carried out at INGV.

---

## Author Comment (AC2) · 5 Jul 2020

Thanks for the detailed review Bernardo Feldman. Your contribute as musician was really appreciated and gave us the chance to greatly improve the paper. Following your suggestions we have extensively reorganized the manuscript.

You wrote: No explanation YET as to how exactly the sonification took place. Did the musicians translate the data by providing their own personal interpretation? Were voltages used to assign amplitudes (loudness/dynamics), frequencies (pitch values), lengths of individual sounds to create rhythmic patterns? There is no clarification given to the reader who might not know what sonification exactly is.

[Figure]

We added a detailed paragraph in which we explain in detail the rules of sonification, based on rigorous scientific criteria, so that any musicians who would follow our methodology would achieve the same pitches collected in the same site. Once the pitches are drawn the musicians can work on them in different ways, as explained in the text.

You wrote: If the intention of the project is to bring awareness to our planet, it really doesn't matter how literal the translation of the data provided by the voltages is. Evidently, as the title of the article clearly states, it is intention to attract people's interest to geoscience while, in the process sensitizing us to planet sustainability. If, in the other hand, the sonification can to be used as a tool to "prevent risks" and assist in the reading of a particular phenomenon occurring at a certain part of our planet, there has to be a consistent array of sounds or textures that could quickly and easily be understood by people, scientists in particular. I would equate these as type of "sonic fingerprints" mentioned in line 145 of the article. I presume that there already many other tools are in place to measure potential earthquakes, eruptions, pollution of aquifers, seawater intrusion along the coastlines, seismic risk, drought and permafrost melting, etc.

We added a paragraph named "Objectives" where we have explained better what we aimed at: we are not interested in any device for preventing risk, but simply to use Music to sensitize people regarding global emergencies. Rather than a single frequency/pitch, we are sure that a cluster of pitches, that can be assigned to a scale and hence to a mood, can arise emotions and curiosity for the listener.

You wrote: What exactly are they looking to find through these experiments in schools? Are they looking to measure how students respond and get sensitized in regards to planet sustainability? Is it by simply bringing attention to the subject?

We focus more this in "Objectives" paragraph

You wrote: Although it is stated that fixed values are used to assign to the various transient's particular pitches in an objective manner, it is not quite clear how these pitches

are arrived at. The analogy comparing it with edges of a color scale does help to assume that gradations have been arrived with a certain logic, most likely translating numerical values into audible frequencies that retain precise intervallic relationships. The explanations right below help in confirm that amplitudes (louder sounds) will come from the stronger responses and vice versa, and that higher frequencies (pitches) will correspond to stronger responses. Inasmuch as I tried to find a clear correspondence between the pitches given in Figure 3, arrived at as transformations of the voltage's responses, and the so-called "score" utilized by the saxophonist to interplay with these collected sonified transients. Although there is no explanation as to how the player choose these particular chords as the basis for his improvisation, while listening to the two elements interacting with each other, I found that the performer did utilize pitches that beautifully harmonized and complemented the clusters created within the sonified textures. While crossfading, alternating pitches within the sonification produced a sense of chromatic movement extending beyond the traditional harmonizations found in "traditional" Western music, and closer to the textures of musicians experimenting with pure intonation and microtonalism.

In "Data sonification" we have reported more details about the sonification rules. Regarding "Selinunte" we corrected a mistake about the grouping of the pitches (it's not by 4). Here is the corrected text:

As the saxophonist Marco Guidolotti played during this reversed part of Selinunte piece, the relative score has the form shown in Figure 4 (notice that the pitches are translated into an Eb instrument, while the sonification produced notes in the C-key). The musician chose to fix some chords that can be assigned by grouping the pitches so to get some chords.

Hence, the saxophonist plays the exact pitches derived from the sonification, but starting from the end of the transient in Fig. 3, so that to reproduce the travel from the greatest depths to the surface. It is important to emphasize that we don't want to keep the musicians in a cage: they can handle the pitches as they wish, in any order: the

most important thing is to remain into the scale and the different moods provided by the local geology.

---

## Editor Comment (EC1) · Isaac Kerlow (Editor) · 7 Jul 2020

I found that the changes incorporated in the update will certainly add clarity to the article, while allowing those of us who aren't either scientists nor musicians to fully participate in its reading. (Typed by Isaac Kerlow, since Dr Feldman was unable to login in).
* * *

---

## Editor Comment (EC2) · Isaac Kerlow (Editor) · 11 Sep 2020

Dear Dr Menghini,

Thank you for replying to the comments provided by Dr Feldman.

We accept your paper with minor comments, please go ahead and upload the revisd version.

Best regards,

Isaac

---

## Author Response (AR2)

We have revised the manuscript by considering the following comments by the Editor:

Lines 25-27
Remove "certainly"
In this trend, the EMusic is certainly the first project that, utilizing the ElectroMagnetic (EM) responses of the Earth, provided a new method to sonify data strictly related to the geological nature of the subsoil.

Line 29
Remove "very"
This concept has a very old origin since

Line 33
Use / instead of \
ferrous\metal

Line 38
- List the reference from which you are quoting since you are quoting a definition from another source
"By definition, it is described as a "direct translation of a data waveform to the audible domain""

Lines 46-48
- Remove "Anyway"
- OPTIONAL: At the end of the sentence perhaps list you top one or two favorite sonification/audification papers so those who are interested can quickly find the info in situ.
Anyway, a detailed overview of sonification in general and the status of the research in this specific field is beyond the scope of this paper so, for an in-depth discussion of sonification and audification concepts, the reader can refer to many other already published papers [SUCH AS].

Line 62
- Use "non-expert audiences" instead of "common people"
- Use "appreciated" instead "understood"
connection between Art (Music) and Science (Geology), in a way that can be easily understood by common people.

Lines 62-63
- Remove "Finally"
- Remove "in all respect"
- I think the start of the sentence:
    "Finally, Geoscientists can be considered in all respect as composers,"
includes a subjective opinion that can also be considered a loaded statement, it needs to be fine-tuned
- Replace with these options or other:
Geoscientists play some of the traditional composers' roles
Geoscientists play many of the traditional composers' roles
Geoscientists actively participate in the composition of the work

Line 86
Add "(Georesonance)" inside parenthesis after "Georisonanze"

Line 93
Use "difference" instead of "differences"

Line 157
Remove quotes, use microtones instead of "microtones"